# Assessment of Ecosystem Services Value in a National Park Pilot

**Xiaodi Zhao [1,2], Youjun He [1,*], Chao Yu [2], Danyun Xu [1] and Wentao Zou [1]**

[1] Research Institute of Forestry Policy and Information, Chinese Academy of Forestry, Beijing 100091, China; zhaoxiaodi@caf.ac.cn (X.Z.); xudanyun121212@126.com (D.X.); zouwt@caf.ac.cn (W.Z.)

[2] School of Economics and Management, Beijing Forestry University, Beijing 100083, China; xuchang@bjfu.edu.cn

\* Correspondence: heyoujun@caf.ac.cn

**Abstract:** Based on the pilot ecosystem analysis of Qianjiangyuan National Park, the ecosystem services function value index system was determined; multiple methods such as market value method and shadow engineering method were used to evaluate the four major categories in the years 2005, 2010, 2015, and 2018 of the ecological services of Qianjiangyuan national park which are provision, regulation, culture, and support. Results show that the total value of the pilot ecosystem services in Qianjiangyuan national park had increased to $7430.11 \times 10^6$ yuan, $9128.41 \times 10^6$ yuan, $12,718.38 \times 10^6$ yuan, and $15,539.99 \times 10^6$ yuan for each category respectively. The regulation category has always been the core ecosystem services function in the national park, accounting for more than 40% of the value of the total services. The increase in the value of ecosystem services in the park was due to the implementation of ecological measures such as logging bans and people paying more attention to environmental protection.

**Keywords:** national park; Qianjiangyuan National Park pilot; ecosystem services; value assessment

## 1. Introduction

Since the establishment of Yellowstone Park in the United States in 1872, the national park construction movement has gradually been promoted in countries around the world. According to IUCN's 2013 Guidelines, "national parks" refer to "large areas of natural or near-nature areas, designed to protect large-scale (large-scale) ecological processes and related species and ecosystem characteristics. However, due to differences in political, economic, and social systems, the specific connotations and functional orientations of national parks vary from country to country. Due to the sparsely populated and rich resources of the United States, the concept of establishing a national park gradually condenses nationality, wilderness (landscape and tourism), history and culture into ecology [1]; the United Kingdom and Japan are subject to a certain degree of human disturbance due to their small size, so it is necessary to protect natural and cultural resources while recognizing human interference with the natural environment.

After more than 60 years of exploration and development, China has gradually established a natural reserve management system with scenic spots and nature reserves as the main body. However, in fact, China's existing natural reserves are incomplete in definition and function. Not fully consistent with internationally certified national parks, such as the nature reserve being the most stringent protection, the scenic spots area emphasizes the tour function; and the classification system and management system of China's protected areas have many problems such as unclear resource ownership and long management heads. In order to solve these problems, China proposed the establishment of a national park system in 2015. Director Zhang Jianlong of the National Forestry and

Prairie Bureau clearly pointed out that taking the road of building a national park according to the actual situation in China and establishing the national park are very clear in the "Overall Plan for Establishing a National Park System". National parks refer to specific terrestrial or marine areas that are approved by the state to establish and lead the management, with clear boundaries, and the protection of large-scale natural ecosystems with national representation, to achieve scientific conservation and rational use of natural resources. Therefore, scientifically assessing the value of ecological resource assets in the pilot zone of the national park system, condensing can achieve a multi-environmental compensation model for the coordinated improvement of ecological functions in concentrated natural conservation areas, sustainable development of the industry, and continuous increase of income of community residents, forming a replicable and scalable methodology The national park construction management technology system is very important.

National park Ecosystem services refer to the environmental conditions and ecological processes that humans depend on for survival and development. It provides food, fresh water, and other raw materials for industrial and agricultural production and, more importantly, maintains Earth's life-support system [2]. However, many of these recreational ecosystem services such as improved health and well-being are difficult to quantify in the model. Additionally, increased numbers of visitors may lead to more resource impact, which can impact all types of ecosystem services. In response to this phenomenon, Taff, Benfield et al. did a study connecting potential impacts from tourism in parks and protected areas to the health and well-being aspect of cultural ecosystem services [3]. Benfield, Nutt, et al. proposed that light pollution in national parks will affect the passenger experience [4]. Miller, Rice et al. studied how to focus on the visitor experience in sustainable tourism, balancing the passenger experience and ecological services [5]. The assessment of the value of ecosystem services in national parks can support national parks to monitor natural resources and, at the same time, formulate national park ecological compensation standards based on the assessment of ecological service values to make up for the lack of ecological compensation in the original nature reserve. On the basis of protection and development, ecotourism product development is carried out based on the results of ecological service value assessment. Therefore, it is important to assess the value assessment of national park ecological services.

In 2000, the state established protected areas in the Qianjiangyuan area, mainly forest parks and nature reserves; in 2015, the state proposed a pilot reform of the national park system, and in June 2016, the Qianjiangyuan National Park pilot was established. In view of the construction process of Qianjiangyuan, this paper makes assumptions about the changes in the ecological service value of Qianjiangyuan National Park in four periods. In 2005, the time for protection in Qianjiangyuan area was still very short, it is assumed that the results of the 2005 ecological service value assessment should be the last; by 2010, the establishment of the Qianjiangyuan protected area had been carried out for 10 years, and the protection effect had been effective. Therefore, it is assumed that the results of the 2010 ecological service value assessment should be significantly improved; before the 2015 Qianjiangyuan National Park pilot was approved, the shortcomings of protected multi-head management emerged, so it is assumed that the results of the Qianjiangyuan ecological service value assessment in 2015 should slow down; after the establishment of Qianjiangyuan National Park in 2018, the construction of national parks improved on the disadvantages of protected areas, so we assume that the results of the Qianjiangyuan ecological service value assessment in 2018 should be the highest. For the ecological services that can be quantified, relevant models are adopted for measurement. Robert Costanza, Rudolf de Groot, et al. provided updated global estimates of ecological services based on updated estimates of unit ecosystem services and land-use changes updated between 1997 and 2011 [6]; Mcbratney, A. B., Morgan, C. L. S et al. estimated the value of ecosystem service contributions by soil [7]; Radford, K.G. James, P studied the changes in the value of ecosystem services along a rural–urban gradient [8]. To provide a scientific basis for the ecological protection and management of Qianjiangyuan National Park, we conducted many years of dynamic monitoring using remote sensing to verify our hypothesis.

## 2. Research Area Overview and Data Sources

### 2.1. Overview of the Qianjiangyuan National Park

The Qianjiangyuan national park is located in Kaihua County, the junction of Zhejiang, Anhui, and Jiangxi (29°10′–29°26′ N; 118°03′–118°21′ E); it is the source of the Qiantang River, the largest river in Zhejiang Province. It has a total area of 252 km$^2$, accounting for 11.27% of the total area of Kaihua. The core protection, ecological conservation, and recreation areas are 72.31 km$^2$, 134.59 km$^2$, and 8.14 km$^2$, respectively. The traditional utilization area is 36.96 km$^2$, involving four towns (Suzhuang, Changhong, Hetian, and Qixi), 19 administrative villages, 72 natural villages, and a population of 9744 people [9]. Qianjiangyuan National Park is in the subtropical monsoon climate zone and exhibits four distinct seasons, abundant precipitation, moderate climate, long frost-free periods, and complex terrain, which constitute a rich and diverse microclimate environment. It has a total annual rainfall of 1963 mm, an average annual temperature of 16.2 °C, a frost-free period of 252 days, and annual average sunshine hours of 1334.1 h [10]. Within the boundaries of the park are Gutianshan National Nature Reserve, Qianjiangyuan National Forest Park, and Qianjiangyuan provincial-level scenic resort. The forest coverage is 81.7% which is relatively high. There are 2230 plant and animal species, and among them are 34 animals under national key protection including black muntjacs (*Muntiacus crinifrons*), white-necked pheasants (*Syrmaticus ellioti*) and Asian black bears (*Ursus thibet-anus*); 32 rare and endangered plant species including Chinese Emmenopterys (*Emmen-opterys henryi*), Hybrid Banana Shrub (*Michelia skinneriana*), and Chinese Stewartia (*Stewartia sinensis*).

### 2.2. Data Sources

Remote sensing data, thematic data, and socioeconomic data were used in this study. Remote sensing data were classified in 2000, 2010, 2015, and 2018. Land use remote sensing monitoring classification data were used in 2000, 2010, while another remote sensing land-use database at a 1:100,000 scale was used in 2015 for the Qianjiangyuan park area. This database was provided by the Institute of Remote Sensing and Digital Earth of the Chinese Academy of Sciences and is based on manual visual interpretation of the Landsat TM/ETM+ 30-meter spatial resolution remote sensing image in 2000, 2010, and 2015. The 2018 land-use remote sensing monitoring classification data is based on a visual interpretation of the Landsat-8 30-meter spatial resolution remote sensing image obtained on 29 November 2018. According to the requirement of Qianjiangyuan Park for the land use classification system, the following land use types in Qianjiangyuan area were identified in the years of 2000, 2010, 2015, and 2018: forest land, shrubbery land, sparse forest land, other forest land, high-coverage grassland, medium-coverage grassland, low-coverage grassland, waters, rural residential land, paddy fields, dry land, and bare land. The level IV land-use classification data was converted into GeoTiff format raster data supported by the InVEST model with a spatial resolution of 30 meters.

Thematic data, including wild animal and plant resources, conservation management fees, and national standards for the protection of animals and plants, is mainly used to calculate the value of rare and endangered species resources. Data of wild animal resources conservation management fee is from the Measures for Collecting Fees for Protection and Administration of Terrestrial Wildlife Resources (No. 72 [1992] of the Ministry of Forestry) and the Protection and Management Fees for Capturing and Hunting National Key Protected Wildlife Resources; national standards of collecting fees for animal and plant protection is taken from the Ministry of Forestry's Notice on How to Determine the Value Standards of National Key Protected Wild Animals and Their Products in Wild Animal Cases (No. 44 [1996] of the Ministry of Forestry). According to the Notice, the value of terrestrial wildlife under first-class State protection is 12.5 times that of the animal resource protection management fee; the value of terrestrial wildlife under second class State protection is 16.7 times that of the animal resource protection management fee.

Socioeconomic data include value, production, and planting areas of different crops, timber management value, forest area, freshwater product value, water supply, tourism costs, and tourist

numbers. Socioeconomic data used in this research was taken from the Statistical Yearbook of National Economic and Social Development of Kaihua County in 2005, 2010, 2015, and 2018; the Water Conservancy of Kaihua County; the statistical weather and climate information from the Kaihua County Meteorological Administration; the Qianjiangyuan National Park pilot Master Plan (2016–2025) provided by the Qianjiangyuan National Park Management Committee; and statistical data from other relevant departments.

## 3. Methods and Improvements

Not all services functions of national parks generate economic value, and sometimes the benefits of certain services are discontinued to obtain other services functions (opportunity costs) [11]. Therefore, defining the type of service for national parks is key to establishing the correlation between the national park's function and value and assessing the ecological function value of national parks [12]. Therefore, based on the classification of MA, this study quantitatively evaluates ecosystem services of the Qianjiangyuan National Park Pilot Project—provisioning services (crop production, timber management, freshwater products, water supply), cultural services (recreational landscape, science/education), regulating services (regulating floods, regulating climate, carbon fixation, and oxygen release), and supporting services (maintaining biodiversity).

### 3.1. Provisioning Services

Provisioning services encompass crop production, timber management, freshwater products, and water supply.

#### 3.1.1. Crop Production

The value of crop production is usually obtained using market value methods. The market value is evaluated by multiplying crop yield and the crop's market price [13]. When studying the crop production value of the park, we also took the cost of crop production into consideration. The calculation is as follows:

$$V_{p1} = \sum A_i \times Q_i \times P_i - C_i \qquad (1)$$

where $V_{P1}$ represents the value of different crops, $A_i$ is the area of different crops, $Q_i$ is the average yield of different crops per unit area, $P_i$ is the market price of crops, and $C_i$ is the production cost of various crops. The main crops in Qianjiangyuan National Park pilot are rice, rapeseed, and tea.

#### 3.1.2. Timber Management

The value of timber is calculated as:

$$V = \sum_{i=1} S_i \times V_i \times P_i \qquad (2)$$

where $S_i$ is the area of type $i$ forest land; $V_i$ is the net growth (m$^3$) of type $i$ forest; $P_i$ is timber price of type $i$ forest (yuan·m$^{-3}$) [14].

Since the area of forest land in the Qianjiangyuan National Park cannot be accurately measured, this paper uses the timber value and forest area of Kaihua County to estimate the value of timber management in the national park. The calculation is as follows:

$$V_1 = V_2 / S_2 \times S \qquad (3)$$

where $V_2$ is the timber value of Kaihua County, $S_2$ is the overall net growth of timber in Kaihua County, and $S$ is the total ecological area of Qianjiangyuan National Park.

### 3.1.3. Freshwater Products

Value of freshwater products is calculated as:

$$V_{p2} = \sum_{i=1}^{n} y_i \times p_i \tag{4}$$

where $V_{p2}$ is the value of freshwater products, $Y_i$ is the yield of the i-th freshwater product, and $P_i$ is the price of the i-th product. Qianjiangyuan pilot area has such freshwater products as fish, crustaceans, shellfish, etc.

### 3.1.4. Water Supply

The calculation for water supply value is:

$$V_{p3} = S(t) \times P_W \tag{5}$$

where $V_{p3}$ is the value of water supply service, S(t) is the quantity of water supply, and $P_w$ is the market price of water.

The wetland in Qianjiangyuan national park is the source of freshwater, so *S(t)* is the water supply of the Qianjiangyuan national park wetland.

### *3.2. Cultural Services*

Cultural services include recreational landscape and science/education.

### 3.2.1. Recreational Landscape

The value of recreational landscape services in ecosystems is usually evaluated using market value methods. Taking the tourist utility level into consideration we used travel cost methods to evaluate the value of the recreational landscape:

$$V_3 = \int_0^{PM} Y(x)dx - V_0 \tag{6}$$

where *PM* is the maximum difference between the fee that a tourist is willing to pay and the fee they actually had to pay, resulting in the consumer utility level; *Y(x)* is the function of travel costs and the number of tourist visits; $V_0$ is the entertainment value of the pilot area in 2018, which is based on growth in consumption value due to waters including boating, overwater dining and water recreations between 2015–2017. According to the statistics of 2015–2017 the number of tourists and tourism expenses in the national park area, the relationship between the increase in tourism expenses and the number of tourists is $Y = Ax^2 + Bx + C$, where Y is the number of tourists and X is the increase in tourism expenses.

### 3.2.2. Science and Education

Using the PRA method, our study conducted semi-structured interviews with researchers and college and middle/high school students to obtain alternative cost and market value estimation according to the number of scientific and educational activities held in the past years and the number of people who benefited, along with scientific and educational results. The value of scientific research and educational ecological service functions was calculated as:

$$V_4 = \sum_{k=0}^{n} x^k \tag{7}$$

where $x^k$ is the cost of recreating scientific research or estimated opportunity cost, $V_4$ is the total value of n research activities.

A large number of scientific research and educational activities were carried out involving Qianjiangyuan National Park; many SCI papers have been published, and Qianjiangyuan means great value to scientific research. Therefore, when evaluating scientific research and cultural/educational services in the Qianjiangyuan National Park, the paper publication is an important part compared with the value of scientific research in other ecological systems (Table 1).

**Table 1.** Evaluation of scientific research and cultural/educational activities.

| Value Type | Content | Method | Calculation |
|---|---|---|---|
| Scientific research value | Basic research | Expenditure method | Estimation of investment in scientific research projects |
| | Application development research | Expenditure method | Estimation of investment in scientific research projects |
| | International research | Expenditure method | Project funding + the actual cost |
| | Teaching/Internship | Alternative cost | Student development fee |
| Cultural value | Thesis | Alternative cost | Student development fee |
| | Publication | Market value method | pages*print copy*price |
| | Video products | Expenditure method | Film investment |
| | Travel expenses | Expenditure method | Transportation, accommodation, and other expenses |

*3.3. Regulating Services*

Regulating services include regulating floods, regulating climate, carbon fixation, and oxygen release.

3.3.1. Regulating Floods

The value of regulating floods:

$$V_{r1} = W_r \times P_c \tag{8}$$

$$W_r = S_r \times R \tag{9}$$

where $V_{r1}$ is the value of regulating floods, $W_r$ is the amount of water storage, $P_c$ is the unit construction cost of the reservoir, $R$ is the unit regulating capacity of the lake, and $S_r$ is the area of the lake.

The wetland ecosystem plays a major role in regulating floods in the park. Therefore, $W_r$ is the amount of water stored in the wetland in the current year, and $S_r$ is the wetland area of that year. The capacity of the wetland reservoir in the park to regulate floods was calculated according to the reservoir storage capacity of Zhejiang Province.

3.3.2. Regulating Climate

The value of regulating climate (humidity) is:

$$V_{r2} = Q_{r2} \times Q_e \times P_{r2} \tag{10}$$

where $V_{r2}$ is the value of regulating humidity, $Q_{r2}$ is the average surface evaporation, $Q_e$ is steam power consumption converted from per unit volume of water, and $P_{r2}$ is electricity price.

The value of regulating climate (temperature) is:

$$V_{r3} = Q_{r3} \times W / Y \times P_{r3} \tag{11}$$

where $V_{r3}$ is the value of regulating humidity, $Q_{r3}$ is the average surface evaporation, $W$ is the heat of vaporization of water, and $P_{r3}$ is the electricity price.

In the pilot area, the wetland ecosystem played a major role in regulating the atmosphere. Therefore, the value of the regulation function was calculated only for the wetland ecosystem; $Q_{r2}$ and $Q_{r3}$ are wetland average surface evaporation. Considering that the service of regulating temperature and humidity only benefits people from May until September in Kaihua County, the service value of humidity and temperature regulation in the wetland is calculated only for this period.

### 3.3.3. Carbon Fixation and Oxygen Release

The value of carbon fixation and oxygen release was evaluated using the INVEST model. The total carbon storage was calculated as follows:

$$C = C_{above} + C_{below} + C_{dead} + C_{soil} \tag{12}$$

where C is carbon storage; $C_{above}$ is above-ground carbon storage; $C_{below}$ is underground carbon storage; $C_{dead}$ is organic carbon storage; $C_{soil}$ is soil carbon storage [15]. Plants can absorb 1.63 t $CO_2$ and release 1.2 t $O_2$ per ton of dry matter yield. The amount of oxygen released is calculated from the amount of carbon sequestered. The value of carbon fixation and oxygen release is calculated according to the total mass, wherein the afforestation cost of $CO_2$ is 1320 yuan/t and the industrial oxygen production cost is 400 yuan/t (Table 2).

**Table 2.** Carbon Storages of Qianjiangyuan National Park in 2000–2018 (Mg C).

|  | Above Ground Carbon Storage | Below Ground Arbon Storage | Soil Carbon Torage | Dead Arbon Torage | Total Arbon Storage |
|---|---|---|---|---|---|
| 2000 | 679,583.76 | 143,540.51 | 3,385,834.28 | 81,552.53 | 4,290,511.02 |
| 2010 | 689,955.89 | 142,466.27 | 3,424,945.94 | 80,661.08 | 4,338,029.13 |
| 2015 | 689,229.89 | 142,329.89 | 3,423,019.12 | 80,630.57 | 4,335,209.41 |
| 2018 | 689,126.02 | 142,301.67 | 3,422,293.54 | 80,710.24 | 4,334,431.42 |

### 3.4. Supporting Services

Supporting services are important for the maintenance of biodiversity. Qianjiangyuan National Park is located in the typical evergreen broad-leaved forest distribution area. The natural vegetation is well preserved and contains abundant rare and endangered plants. It is an important area for the distribution of rare and endangered plant species in Zhejiang Province. Therefore, the value of rare and endangered species instead of a biodiversity value is used. For example, there has National level key protected wild plant Taxus wallichiana var. maire, National secondary protected wild plants Torreya grandis, China's unique national level I key protected animals and threatened species in the world Syrmaticus ellioti etc.

Biodiversity

Market value methods are used to assess the value of biodiversity. Qianjiangyuan national park has a wide range of biological resources including various rare and endangered animal and plant species that account for a large proportion of the biodiversity value. Therefore, the value of rare and endangered species instead of a biodiversity value is used [16]. The value is calculated as:

$$P_f = \sum_{i=1}^{n} P_i \times M_i \tag{13}$$

$$S_f = \sum_{i,j=1}^{n} R_{ij} \tag{14}$$

where $P_f$ is the total value of rare and endangered animal and plant resources, $S_f$ is the total ecosystem services value of rare and endangered plant and animal resources, and $P_i$ is the conservation management fee of the i-th rare and endangered animal and plant resource, $M_i$ is the value of the multiplier of the i-th rare and endangered animal and plant resource, $R_{ij}$ is the j-type ecosystem service value of the i-th rare and endangered animal and plant resource. When studying the biodiversity value of Qianjiangyuan national park, we assumed that the ecosystem services value of rare and endangered animal and plant resources is the total value of rare and endangered animal and plant resources. Therefore:

$$S_f = P_f \tag{15}$$

In summary, this research uses the Millennium Ecosystem Assessment (MA) classification method and takes the characteristics of the ecosystem of the Qianjiangyuan national park pilot project into consideration. As a result, two different types of ecosystem services values can be distinguished direct and indirect values. There are four types of services functions that we identified—provisioning, cultural, supporting, and regulating function, establishing the wetland ecosystem service value index system in the Qianjiangyuan national park pilot project (Table 3).

**Table 3.** Ecosystem service value index system in Qianjiangyuan National Park pilot.

| Value Type | Services Type | Evaluation Index | Evaluation Method |
|---|---|---|---|
| Direct value | Provisioning services | Timber management | Market value method |
| | | Crop production | Market value method |
| | | Freshwater products | Market value method |
| | | Water supply | Market value method |
| | Cultural services | Recreational landscape | Travel cost method |
| | | Science and education | Alternative cost method |
| | Regulating services | Regulating floods | Shadow engineering method |
| | | Regulating climate | Shadow engineering method |
| Indirect value | | Carbon fixation and oxygen release | INVEST model |
| | | Maintaining biodiversity | Market value method |
| | Supporting services | Providing species habitat | Market value method |

## 4. Results and Analysis

### 4.1. Change in Value of Provisioning Services

#### 4.1.1. Value of Crop Production

Rice is the main food crop in Kaihua County. The paddy rice planting area in the Qianjiangyuan national park is 20,719.95 mu, accounting for 18.84% of the Kaihua County, and is mainly divided into early-and single-season late rice, and continuous cropping late rice. Rapeseed is the main oil crop, and tea is another income source for residents. Using the formula, we determined that (1), the value of food crops is obtained as 0.20 billion RMB, 0.34 billion RMB, 0.46 billion RMB, and 0.45 billion RMB for the years 2005, 2010, 2015, and 2018, respectively (Table 4). From 2005 until 2018, the overall value of crops increased first and then decreased later. Changes in land use showed that the planting area of crops decreased in varying degrees between 2005 and 2016. However, productivity has improved significantly since 2010 due to the development of science and technology. As a result, high-yield rice was widely planted, and the yield of rapeseed and tea was also significantly improved. Rice production in 2015 increased by 12,633 tons compared with that in 2010, resulting in a significant increase in the value of food crop services from 2010; with the development of the Qianjiangyuan National Park, more arable land is protected from destruction and the planting area of crops is smaller, which means that residents mostly rely on tea picking and working in cities. Therefore, the value of rice production in 2018 decreased by 53.93 million yuan compared with 2015, while the value of tea production increased.

**Table 4.** Value of ecosystem services in Qianjiangyuan National Park in 2005, 2010, 2015, and 2018.

| Types of Ecosystem Services | 2005 | 2010 | 2015 | 2018 |
|---|---|---|---|---|
| Provisioning services ($\times 10^6$ Yuan) | 321.91 | 409.76 | 544.1 | 530.36 |
| Crop production | 197 | 339 | 457 | 445 |
| Timber management | 110.81 | 41.65 | 27.28 | 17.86 |
| Freshwater products | 13.9 | 28.7 | 59.2 | 66.6 |
| Water supply | 0.21 | 0.36 | 0.6 | 0.86 |
| Cultural services ($\times 10^6$ Yuan) | 543.37 | 1560.67 | 4788.83 | 7477.3 |
| Recreational landscape | 395.3 | 1316.36 | 4386.45 | 6707.3 |
| Scientific research and education | 148.07 | 244.31 | 402.38 | 770 |
| Regulating services ($\times 10^6$ Yuan) | 5665.23 | 6348.38 | 6575.85 | 6722.73 |
| Regulating floods | 5.85 | 7.55 | 18.5 | 45.33 |
| Regulating atmosphere | 510.77 | 411.61 | 274.41 | 182.3 |
| Carbon fixation and oxygen release | 5148.61 | 5929.22 | 6282.94 | 6294.1 |
| Supporting service ($\times 10^6$ Yuan) | 809.6 | 809.6 | 809.6 | 809.6 |
| Maintaining biodiversity | 809.6 | 809.6 | 809.6 | 809.6 |
| The total value of ecosystem services ($\times 10^6$ Yuan) | 7340.11 | 9128.41 | 12718.38 | 15539.99 |

### 4.1.2. Value of Timber Management

Timber production in the national park mainly includes plantation of forests, forest products, the harvest of bamboo and wood, and collection of wild plants. Using our formula (3), the value of timber management was calculated as 0.11 billion RMB, 0.04 billion RMB, 0.03 billion RMB, 0.02 billion RMB for 2005, 2010, 2015, and 2018, respectively (Table 4). The value of timber management decreased during 2005–2018. The possible reason for this is the decline in logging-based income and the value of local timber management, which is due to policies of returning farmland to forests and protecting forests in the pilot area.

### 4.1.3. Value of Freshwater Products

As a source of the Qiantang River, the Qianjiangyuan national park has rich fish resources, including crustaceans, shellfish among other species. A total of 83 fish species in the park belong to 9 orders and 15 families within the animal kingdom. Among these, grass carp has the highest yield, normally accounting for 50% of the total production, followed by crucian carp, silver carp, and bighead carp. The main aquatic animals include frogs, snails, clams, and shrimps, which have a habitat in various types of waters. Among these, crabs and giant river prawns (*Macrobrachium rosenbergii*) are the main freshwater crustaceans [17]. Using the formula (4), the value of aquatic products in the park was calculated as 0.01 billion RMB, 0.03 billion RMB, 0.06 billion RMB, and 0.07 billion RMB, in 2005, 2010, 2015, and 2018, respectively (Table 4). Due to the new reservoir construction and increased reservoir area in Kaihua County, freshwater production increased during 2005–2018. Moreover, in 2015 China decided to establish a national park system, and as a consequence, the water source environment in the Qianjiangyuan National Park improved significantly, which made the protection and restoration of a large number of freshwater organisms possible and contributed to the increase in freshwater production and its value.

### 4.1.4. Value of Water Supply

Wetland in the national park is a source of freshwater, which is vital for human survival and development. The water source for residents in the pilot area mainly comes from the reservoirs which supply water to four towns, Suzhuang, Changhong, Hetian, and Qixi, accounting for 2.7% of the County's water supply [18]. Using formula (5), the value of water supply in the Qianjiangyuan national park pilot was calculated as 210 thousand RMB, 360 thousand RMB, 600 thousand RMB, and 860 thousand RMB respectively (Table 4). From that it can be seen that the value of water supply

has increased between 2005 and 2018, however, the growth slowed down over the years. A possible reason is that the population in the pilot area and the number of migrant workers increased between 2005 and 2018. This resulted in less demand for water resources in the pilot area and less increase in water supply and its value.

### 4.2. Change in Value of Cultural Services

#### 4.2.1. Recreational Landscape

With ecological protection, the plans of the Qianjiangyuan National Park and the development of its core functions such as community development, recreational landscape, and scientific education developed. The value of the recreational landscape was calculated using the formula (6) as 0.40 billion RMB, 1.32 billion RMB, 4.39 billion RMB, and 6.71 billion RMB (Table 4). As can be seen, the value increased from 2005 to 2018, mainly due to the establishment and continuous improvement of the park, and the governmental support of the local tourism industry.

#### 4.2.2. Science and Education

The Qianjiangyuan national park pilot is an ecological barrier and an important water source conservation in Zhejiang Province and East China. The valleys and wetlands in the area are also important habitats for wild animals and birds. At the same time, a large number of studies and education have been conducted, and some SCI papers were published, showing the park's great scientific value. Using the formula (7), the value in science and education in 2005, 2010, 2015, and 2018 was obtained—0.15 billion RMB, 0.24 billion RMB, 0.40 billion RMB, and 0.77 billion RMB (Table 4). An increasing trend during 2005–2018 can be seen from these values. The possible reason is that with the advancement of science and technology, the demand for land use in various areas of Qianjiangyuan national park has increased, and the demand for research on the rich animal and plant resources has increased. Moreover, the development of transportation and further popularization of relevant knowledge help to promote the value of scientific research and culture and education.

### 4.3. Change in Value of Regulating Services

#### 4.3.1. Value of regulating floods

Wetlands playa a major role in regulating flooding in national parks are one of the core values functions of wetland ecosystems. The value of regulating floods is calculated using formula (8) as $5.85 \times 10^6$ Yuan, $7.55 \times 10^6$ Yuan, $18.5 \times 10^6$ Yuan, $45.33 \times 10^6$ Yuan in 2005, 2010, 2015, and 2018, respectively (Table 4). The value showed an increasing trend during 2005–2018, with an average annual increase of $3.04 \times 10^6$ Yuan. The main reason is that Kaihua County has strengthened its construction of water channels to further improve its water conservation level. In order to achieve the first-level drought tolerance capacity, the area of the reservoir in the Qianjiangyuan national park has increased during 2005–2018. Thus, the value of regulating floods has increased accordingly.

#### 4.3.2. Value of Regulating Climate

To regulate the climate mainly means to regulate atmospheric humidity and temperature. Using the formula (9), the value of regulating climate was calculated as 0.51 billion RMB, 0.41 billion RMB, 0.27 billion RMB, 0.18 billion RMB in 2005, 2010, and 2015 and 2018 respectively (Table 4). The downward trend during 2005–2018 has two main reasons: (1) due to an increased population there has been a wider demand for land use, which has resulted in a decrease in wetland area and a decrease in circulation between wetland and the atmosphere, which weakens the regulation of climate by wetlands as a consequence; (2) global climate change has caused global temperature increase and precipitation decrease. The average annual water surface evaporation has decreased between 2005 and 2018, and the vegetation reduction has further reduced the regulation of climate by wetlands.

### 4.3.3. Value of Carbon Fixation and Oxygen Release

Through literature review, this paper determines the annual carbon sequestration per unit area for forest land, sparse forest land, other forest land, and shrubbery land in Qianjiangyuan area. Increased carbon sequestration from forest growth was also calculated and the calculation results from the InVEST carbon model were modified, and the total carbon storage was obtained for the years monitored. Finally, the value of carbon sequestration and oxygen release was calculated using the carbon sequestration price of 1200 yuan/t with reference to the Swedish carbon tax rate. Using formula (10), the value of carbon sequestration in 2000, 2010, 2015, and 2018 was 5.15 billion RMB, 5.93 billion RMB, 6.28 billion RMB, and 6.50 billion RMB respectively (Table 4). As can be seen, the value of carbon sequestration and oxygen release in 2005–2018 has increased. Two reasons can explain this increase: (1) there has been little variation in land use types in the Qianjiangyuan park area for the past 20 years; (2) increase in carbon sequestration from the annual growth of forest lands, sparse forest lands, other forest lands and shrubbery lands has been ignored during calculation using the InVEST carbon model.

### 4.4. Change in Value of Supporting Services

Value of Biodiversity

The value of biodiversity is the major part of the value of supporting services. Using formula (14), the value of biodiversity was calculated as 0.81 billion RMB for 2005, 2010, 2015, and 2018 (Table 4). Since our study uses rare and endangered wild animals and plants to measure the value of biodiversity, it is assumed that the value of biodiversity remains the same across the years monitored.

### 4.5. Change in Total Value

Using market value, alternative cost, shadow engineering and other evaluation methods, this study evaluated the ten major ecosystem services in the Qianjiangyuan Park. As shown in Table 4, the total value of ecosystem services in Qianjiangyuan is 7.34 billion RMB, 9.13 billion RMB, 12.72 billion RMB, and 15.54 billion RMB for 2005, 2010, 2015, and 2018 respectively, showing an increasing trend. The national park has brought huge benefits to Kaihua County. The assessment results do not only bring more attention from the public and the government to national parks, but also provides bases for rational allocation of national parks in regional planning.

As can be seen in Figure 1, among the ten ecosystem services functions assessed, regulating services has the highest value out of the four major categories of ecosystem services (except for 2018 when cultural services had higher value than regulating services), accounting for 77.18%, 69.55%, 51.70%, and 43.26% of the total in 2005, 2010, 2015, and 2018, respectively.

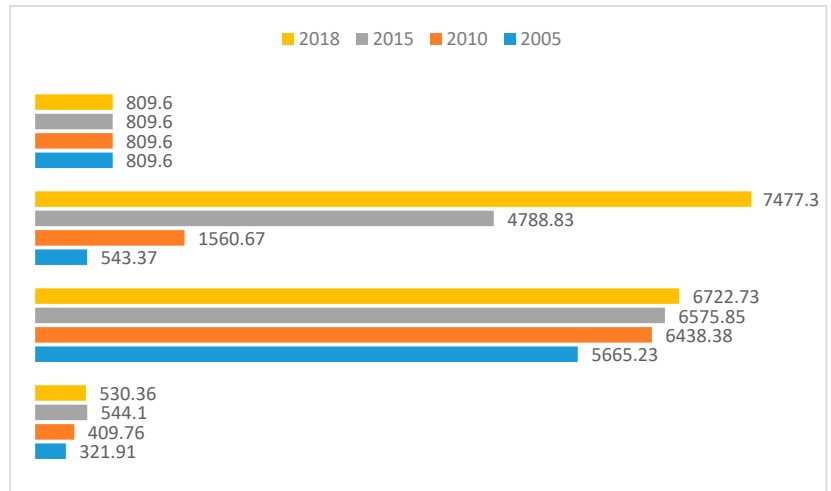

**Figure 1.** Evaluation of different ecosystem services in 2005, 2010, 2015, and 2018.

Figure 1 shows that the value of provisioning services first increased. Compared with 2015, it then decreased by $13.74 \times 10^6$ Yuan in 2018. Similarly, the value of crop production increased first and then decreased; the value of timber management showed a decreasing trend overall; the value of freshwater products and water supply increased. The overall value of cultural services showed an increasing trend, among which the value of the recreational landscape, science and education showed an increasing trend too. The value of regulating services showed an increasing trend overall, among which the value of regulating floods and carbon sequestration and oxygen release has increased, and the value of regulating climate has decreased

## 5. Conclusion and Discussion

The overall value of ecosystem services in the park has increased across the years, which is mainly because the value of cultural and regulating services has increased, while the value of supporting services was unchanged, and the value of provisioning services decreased after an initial increase. As the higher value of cultural and regulating services was greater than the decrease in the value of provisioning, the total value of ecosystem services has increased across the years.

The value of cultural services function in the Qianjiangyuan national park has increased across the years. Our study identified the following reasons for this increase: (1) increase in the value of recreational landscape due to the establishment of a local tourism industry; (2) with the advancement of science and technology, the value of various areas have been further explored, and the popularization of relevant knowledge brought up by progress in transportation has led to an increase in the value of education and scientific research. As the value of recreational landscape and scientific research increases every year, the value of cultural services function increases accordingly.

The value of regulating services function has increased across the years. The main reasons for this increase might be: (1) the construction of reservoirs in Kaihua County has increased the water conservation capacity, and the value of regulating and storing floods has increased accordingly; (2) reduction in wetland area leads to a decrease in the value of regulating climate; (3) the value of carbon sequestration and oxygen release has increased across the years. Since the increased value of carbon sequestration and oxygen release and regulating floods is greater than the decrease in the value of regulating climate, the value of regulating services increased across the years.

The value of provisioning services function has first increased and then decreased. The main reasons for this trend may be: (1) the return of farmlands to forests and protecting forests implemented in the pilot areas. The value of timber management and logging-based income has declined; (2) with the development of national parks, more arable land is protected, the planting area of crops is reduced, and residents mostly rely on tea-picking and working in cities, and the value of crop production declines. Although the value of freshwater products and water supply increased across the years, the increased amount was less than the reduction in the value of crops and timber management. Therefore, the value of provisioning services function increased first and then decreased.

The value of supporting services function remained unchanged. The main reason is that our study uses rare and endangered wild animals and plants to measure the value of biodiversity. Therefore, it is assumed that the value of biodiversity remains the same across the years.

According to the evaluation of the ecological service value of Qianjiangyuan National Park, the evaluation results of the ecological service value in 2005 were the lowest, the results in 2010 increased significantly, and the results in 2018 increased compared with the previous ones, consistent with the assumptions presented in this paper. Only the results of the 2015 ecological service value assessment did not reduce the growth rate. This paper argues that the substantial increase in the value of cultural services in the Qianjiangyuan area in 2015 has increased the overall assessment of ecological services.

Studies of changes in the value of ecological services under land use change in the Qianjiangyuan national park pilot was conducted by Chinese scholars. For example, Sun et al. (2019) designed four land use change scenarios in 2025, business as usual, strategic planning, ecological protection, and development, and analyzed ecosystem services and its value change in Qianjiangyuan National

Park, water supply, water conservation, carbon sequestration and oxygen release, soil conservation, environmental purification, and habitat quality, using InVEST model and CLUE-S model. Results show that from the perspective of ecosystem service types, the value of carbon sequestration and oxygen release was highest, which is in line with our research results; from the perspective of ecological services function, the value of ecosystem services in core protected areas and ecological conservation areas was the highest; from the perspective of land use scenario, the value of ecosystem services in the ecological protection scenario was the highest, and was the second highest in the strategic planning scenario; the water supply service in the strategic planning scenario was better than that in the ecological protection scenario, while other ecosystem services were inferior to that in the ecological protection scenario. Considering that the Qianjiangyuan national park provides water resources for downstream areas, Sun regarded the strategic planning scenario as the optimal land use strategy in 2025.

Compared with their research on the value of ecological services in national parks under different land use scenarios, our research on the value of ecological services was based on market values, where carbon sequestration and oxygen release were taken into consideration based on land use changes. On the other hand, Sun only considered water supply in the provisioning services function, soil and water conservation, environmental purification, and carbon sequestration and oxygen release in the regulating services function, and habitat quality in the habitat services function. Our study includes a more comprehensive assessment of these ecosystem services functions including provisioning services, cultural services, regulating services, and supporting services.

Nonetheless, further studies are required to evaluate the complex connections between the Qianjiangyuan national park and the surrounding urban entities; other benefits of ecological services function in the park such as increasing the value of the surrounding real estate market and promoting the city image were not considered, resulting in a calculated value of ecosystem services for the Qianjiangyuan national park that is lower than in reality. The next step is to consider the connections between the national park and surrounding communities. National parks are not only vital to biodiversity conservation but are also vital to many local residents who depend on natural resources for their survival. In the 10 national park system pilot areas established in China, there are Aboriginal life and community issues that have become one of the unavoidable problems in the construction of national parks. Therefore, in the early stage of the establishment of Qianjiangyuan National Park, the research community participation mechanism has reference significance for institution building work. Qianjiangyuan National Park should protect the natural and cultural resources through the construction of community participation mechanisms and realize the positive interaction between the community and the national park.

**Author Contributions:** Conceptualization, X.Z. and Y.H.; Methodology, X.Z. and W.Z. Formal Analysis, C.Y., D.X.; Investigation, C.Y. and W.Z. Writing-Original Draft Preparation: X.Z.

**Funding:** This research was funded by Fundamental Research Funds for Youth Research Projects at the Central Level Nonprofit Research Institute "Study on Ecological Compensation and Management System in the National Park System Pilot Area" grant number CAFYBB2017QC006.

**Acknowledgments:** Thanks to the Qianjiangyuan National Park Management Commission for the data support.

**Conflicts of Interest:** We declare no conflict of interest.

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
