# Peer review of "Assessment of Ecosystem Services Value in a National Park Pilot"

_sustainability, doi:10.3390/su11236609_

Round 1
Reviewer 1 Report
Overall, this is a fairly well-written paper and will make a good contribution to the literature pending minor revisions. I recommend the following revisions before publication.
The introduction is not sound and needs to be revised. The authors discuss national parks and compare them to the US system, but their comparison is not correct. National Parks were originally established for scenic and tourism values, not for ecological reasons. Additionally, the idea of national parks started somewhat in Yosemite Valley, not necessarily Yellowstone. Lastly, the idea of natural resource development in many non-Chinese national parks is antithetical to the idea of national parks, the very idea is opposed to development. See the following resources: Runte, A. (1997). National parks: the American experience. U of Nebraska Press. Miller, Z.D., Rice, W., Taff, B.D., & Newman, P. (2019). Concepts for understanding the visitor experience in sustainable tourism. In S.F. McCool and K. Bosak (Eds.), A Research Agenda for Sustainable Tourism.Cheltenham, UK: Edward Elgar Publishing. The authors must recognize in their limitations that the ecosystem services they are identifying are very limited and not exhaustive. This is particularly true for recreation ecosystem services (under cultural). Many of these recreational ecosystem services are difficult to quantify and not included in this model. Such as: improved health and well-being, increased spiritual connection, restoration, etc. etc. Additionally, the authors should acknowledge that there are inherent tradeoffs between these services: increased numbers of visitors may lead to more resource impacts, which can impact all types of ecosystem services. Authors should work to incorporate the following citations as part of their work: Miller, Z.D., Rice, W., Taff, B.D., & Newman, P. (2019). Concepts for understanding the visitor experience in sustainable tourism. In S.F. McCool and K. Bosak (Eds.), A Research Agenda for Sustainable Tourism.Cheltenham, UK: Edward Elgar Publishing.
Taff, D., Benfield, J., Miller, Z.D., Schwartz, F., & D’Antonio, A. (2019). The Role of Tourism Impacts on Cultural Ecosystem Services. Environments, 64(4), 43.
Benfield, J., Nutt, R., Taff, D.,Miller, Z.D., Costigan, H., & Newman, P. (2018). A laboratory study of the psychological impact of light pollution in national parks. Journal of Environmental Psychology. 57, 67-72.
Reviewer 2 Report
This is a very interesting paper. There are a few changes that could be made to allow for the paper to be more easily read:
The introduction needs to be edited. There is a lot of useful facts in the introduction, but they are not presented in a way that is easy to follow. Perhaps you could start with Catlin's 1832 proposal, then discuss the Yellow Stone and China adoption. This could then be followed by the multiple reasons it has been implemented in the U.S. and China. Be sure to introduce acronyms before using (e.g. SCI) When describing large sums of money, it is easier to read 7.34 billion than 7340.11 x 106 For Section 3.4, it would be useful to describe what these supporting services are and provide examples. In the conclusion, you mention that the connection between the national park and surrounding communities. It would be useful to expand on this idea and why this is the next logical step.
